# Estimating Individual Conifer Seedling Height Using Drone-Based Image Point Clouds

**Guillermo Castilla [1],\* , Michelle Filiatrault [1] , Gregory J. McDermid [2] and Michael Gartrell [1]**

1   Canadian Forest Service, Natural Resources Canada, 5320 122 Street Northwest, Edmonton, AB T6H 3S5, Canada; michelle.filiatrault@canada.ca (M.F.); michael.gartrell@canada.ca (M.G.)
2   Department of Geography, University of Calgary, Calgary, AB T2N 1N4, Canada; mcdermid@ucalgary.ca
\*   Correspondence: guillermo.castilla@canada.ca; Tel.: +1-(825)-510-1186

**Abstract:** *Research Highlights:* This is the most comprehensive analysis to date of the accuracy of height estimates for individual conifer seedlings derived from drone-based image point clouds (DIPCs). We provide insights into the effects on accuracy of ground sampling distance (GSD), phenology, ground determination method, seedling size, and more. *Background and Objectives:* Regeneration success in disturbed forests involves costly ground surveys of tree seedlings exceeding a minimum height. Here we assess the accuracy with which conifer seedling height can be estimated using drones, and how height errors translate into counting errors in stocking surveys. *Materials and Methods:* We compared height estimates derived from DIPCs of different GSD (0.35 cm, 0.75 cm, and 3 cm), phenological state (leaf-on and leaf-off), and ground determination method (based on either the DIPC itself or an ancillary digital terrain model). Each set of height estimates came from data acquired in up to three linear disturbances in the boreal forest of Alberta, Canada, and included 22 to 189 surveyed seedlings, which were split into two height strata to assess two survey scenarios. *Results:* The best result (root mean square error (RMSE) = 24 cm; bias = −11 cm; $R^2$ = 0.63; $n$ = 48) was achieved for seedlings >30 cm with 0.35 cm GSD in leaf-off conditions and ground elevation from the DIPC. The second-best result had the same GSD and ground method but was leaf-on and not significantly different from the first. Results for seedlings ≤30 cm were unreliable (nil $R^2$). Height estimates derived from manual softcopy interpretation were similar to the corresponding DIPC results. Height estimation errors hardly affected seedling counting errors (best balance was 8% omission and 6% commission). Accuracy and correlation were stronger at finer GSDs and improved with seedling size. *Conclusions:* Millimetric (GSD <1 cm) DIPC can be used for estimating the height of individual conifer seedlings taller than 30 cm.

**Keywords:** drone-based image point clouds (DIPC); Unmanned Aerial Vehichles (UAV); photogrammetry; forest monitoring; forest inventory; restoration

## 1. Introduction

The boreal forest of Alberta, Canada, is a busy landscape where widespread industrial activity (oil and gas, mining, and forestry) fragments and negatively affects the habitat of threatened species such as woodland caribou (*Rangifer tarandus caribou*). In an effort to mitigate this impact and avoid temporary prohibitions of industrial activity stemming from species at risk legislation, industry and government alike are investing in restoring anthropogenic disturbances, in particular seismic lines (narrow geophysical survey corridors), which have been linked to the decline of woodland caribou [1,2].

The Alberta seismic line restoration framework includes two surveys to monitor regeneration/ restoration success: the survey survey between years 2 and 5 after treatment, and the establishment survey between years 8 and 10 [3]. The first survey estimates the survival of seedlings from acceptable

tree species, and the second determines whether the surveyed area meets the establishment targets in terms of overall stocking and spatial distribution of those seedlings. Both surveys have height thresholds below which a seedling does not count. For conifer seedlings, the threshold is 15 cm in the survival survey and 60 cm in the establishment survey. Hence, height is an important factor not only in the assessment of recovery trajectories, but also in the decision of whether a seedling should be counted. The goal of this paper is to assess the degree to which it is possible to estimate the height of conifer seedlings growing along recovering linear disturbances using drone-based image point clouds (DIPCs). We consider two scenarios: the survival assessment survey, where the seedlings are very small, and the establishment survey, where most seedlings will exceed 30 cm height. We focus on conifer seedlings because they are preferred over broadleaf tree species for restoring mixedwood boreal forests, since they usually represent the late-stage succession path of these forests. If drones (also known in the literature as unmanned aerial vehicles or remotely piloted aircraft systems, UAV or RPAS respectively) could be used to detect seedlings and count those exceeding the height requirement, this would greatly facilitate the surveying of regenerating linear disturbances quickly and efficiently. Our previous experience with detecting seedlings with drones [4,5] suggests that there is still significant research to be carried out before fully automated drone surveys, which should include other attributes such as species and height, become operational. In this paper we focus on height estimation and aim to provide insights about the following research questions:

1. How accurately can conifer seedling height be estimated in drone surveys for survival (small seedlings) and establishment (older and taller seedlings)?
2. Does the accuracy of DIPC differ much from that of softcopy (manual) photogrammetry?
3. What is the impact of height-estimation error on counting errors in seedling surveys?
4. What is the effect of spatial resolution, season, ground determination method, and seedling size on the accuracy of seedling height estimates?
5. What factors contribute to outliers?

Some related questions have been investigated by other authors, although in different contexts. The first studies on the use of digital aerial photogrammetry (DAP) to measure tree height in forest environments were published in the early 2010s [6–9] and concluded that this technique is less accurate than airborne laser scanning (ALS). However, these studies were based on aerial imagery of decimetric resolution (ground sampling distance (GSD) >10 cm), which is considerably coarser than the resolution that drones can achieve. Although ALS may be more accurate for counting mature trees, Chen et al. [10] pointed out that ALS datasets with densities high enough to characterize short vegetation would be too expensive; thus, they recommended DIPC as an alternative, although not with that name. (Note that the acronym DIPC was coined by Kotivuori et al. [11]; we prefer DIPC over DAP because it is more specific. For example, manual tree height measurements from stereo softcopy would still qualify as DAP, but not as DIPC.) Chen et al. [10] surveyed vegetation height on seismic lines in Alberta's boreal forest using a point-intercept sampling strategy and compared ground measurements with height estimates from DIPC data derived from imagery of 1.2 cm GSD. They concluded that this technique could replace traditional ground surveys at the site level [10].

The first peer-reviewed paper that included seedling height estimation in a forest regeneration context was published in 2017 by Goodbody et al. [12]. They field-measured the height of a representative young tree in each of the 72 circular plots (3.99-m radius) they studied in regenerating harvest blocks in British Columbia. The measured heights ranged from 0.5 to 6 m. They compared that height with the height of the tallest point within the plot DIPC, which was derived from imagery of 2.4 cm GSD, and obtained a root mean square error (RMSE) of 91 cm and a coefficient of determination ($R^2$) of 0.83. A year later, Röder et al. [13] went a step further by studying individual young Norway spruce (*Picea abies* (L.) Karst) in eight 762-m$^2$ plots (mean tree height 1.2–4.1 m) in a conservation area in southern Germany. They obtained a mean RMSE of 1.57 m and an $R^2$ of 0.74 when comparing the DIPC-derived height (from 5 cm GSD imagery) of detected young trees with their field-measured

height. In 2019, in a study of 580 circular plots (50 m$^2$ size and mean tree height 2.5 m) in regenerating stands in Norway, Puliti et al. [14] obtained an RMSE of 77 cm and an $R^2$ of 0.52. They estimated the mean tree height of these plots using a random forest model that had predictors derived from a 3 cm GSD DIPC dataset. Theirs was the first study to demonstrate that DIPC data can provide more accurate predictions than ALS (the RMSE of the best ALS model was 80 cm, even though the ALS data had a density of 5 points/m$^2$). Also that year, Imangholiloo et al. [15] estimated the mean height of small trees within 15 plots in regenerating conifer-dominated stands in Finland (mean height 1.1–3.3 m) using 2.5 cm GSD DIPCs (both leaf off and leaf on). They obtained an RMSE of 52 cm and an $R^2$ of 0.97 with the leaf-off DIPC, and an RMSE of 27 cm and an $R^2$ of 0.95 with the leaf-on DIPC. None of these studies used a millimetric (<1 cm) GSD, and none dealt with seedlings with a mean height less than 1 m.

## 2. Materials

### 2.1. Study Area

This study was carried out in the boreal forest natural region of northeastern Alberta, Canada [16]. The study area is approximately 180 km south of Fort McMurray and consists of a mosaic of upland and wetland vegetation communities on gently undulating, previously glaciated terrain (Figure 1). Two-thirds of the forests in the area are coniferous, ranging from wet, low-lying stands of black spruce (*Picea mariana*) intermixed with larch (*Larix laricina* (Du Roi) K.Koch), to drier upland sites dominated by jack pine (*Pinus banksiana* Lamb.). Broadleaf and mixed forests include trembling aspen (*Populus tremuloides Michx.*) and balsam poplar (*Populus balsamifera* L.). The mean annual precipitation in the region is 440 mm, with a mean monthly temperature ranging from −17 °C in winter to 16 °C in summer [17]. This area overlays a rich bitumen deposit that has been the subject of intense oil and gas activity in recent decades, resulting in a network of well sites, pipelines, and seismic lines that fragment the landscape.

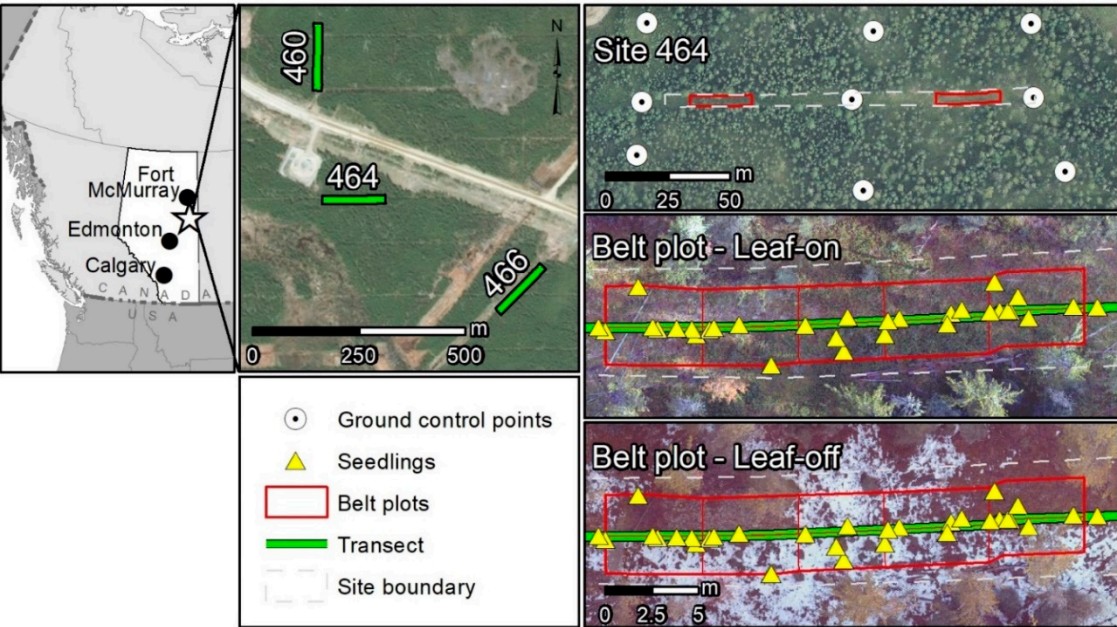

**Figure 1.** Left: study area location in Alberta, Canada. Center: Close-up of the study area showing the three sites. Upper right: site layout. Middle right: example of drone-based image point cloud (DIPC) orthomosaic of a belt plot in leaf-on conditions. Lower right: Same in leaf-off conditions. NB. The transect line divides the belt plot in two halves, hence there are ten sub-plots, five at each side.

Three sites were selected for this study among those established in the summer of 2017 by the Boreal Ecosystem Recovery Assessment (BERA, www.bera-project.org) project, which in addition to seedling monitoring research, included investigations of other aspects of restoration such as coarse woody debris [18], canopy openings [19], and groundwater [20]. The selected sites are two linear disturbances resulting from seismic lines (sites 460 and 464) and an abandoned pipeline corridor (site 466). The sites have probably been inactive since the early 2000s and show various degrees of natural regeneration and a variety of seedling heights. Site 460 has a north to south orientation with a mean slope of 6%. The site understory is dominated by Labrador tea, rose bushes, and graminoids, and the ground is covered with moss and lichen. Site 464 has an east to west orientation with a mean slope of 6%. The site has a thicker moss ground cover than site 460 and has fewer graminoids. The understory shrub cover is similar to that of site 460; however, there is a larger presence of trembling aspen and balsam poplar. Site 466 has a southwest to northeast orientation gently sloping down to the southwest. The site is the driest of the three sites, with a mean slope of 5%, and runs parallel to a high-voltage power line. The understory is dominated by Labrador tea with taller shrubs consisting of willow, dogwood, and rose. Young trembling aspen and balsam poplar trees are scattered throughout the site and adjacent areas.

## 2.2. Field Data

Each site is a 150 m-long section of linear disturbance, within which two 100 $m^2$ belt plots 75 m apart were established at opposite ends of the site. Each belt plot was further subdivided into ten 10-$m^2$ subplots. The species, height, and precise coordinates of all conifer seedlings intersecting the 150 m long site centerline were recorded, as well as those from the tallest representative seedling within each belt subplot (Figure 1).

There were a total of 189 conifer seedlings surveyed in the three sites selected for this study, although not of all them are included in some of the DIPC datasets. For example, only one of the two belt plots at each site was flown at the finest GSD, meaning only 69 seedlings have 0.35 GSD data. All seedlings exceeded the minimum height required for the survival assessment (15 cm), 54 were small (≤30 cm), and none exceeded the "green-up" height of 3 m (Table 1). At site 460, 34 conifer seedlings were measured: 94% black spruce and the rest larch. Out of the 34, only 8 were captured in the 0.35 cm GSD dataset. Eighty-four conifer seedlings were measured at site 464: 63% black spruce, 29% larch, and the remaining jack pine. Out of the 84, 36 were captured in the 0.35 cm GSD dataset. There were 71 conifer seedlings measured at site 466: 92% were black spruce, and the remaining were distributed evenly between larch and jack pine. Out of the 71, 25 were captured in the 0.35 cm GSD dataset (Table 1).

**Table 1.** Drone acquisition parameters and descriptive statistics of ground-measured heights for seedlings used in the analysis of each drone dataset. GSD = ground sampling distance, Sb = black spruce, Pj = jack pine, Lt = larch. NB. The different number of seedlings in the 0.75 cm GSD leaf-on and leaf-off acquisitions, and the different minimum height in the 0.35 cm GSD leaf-on acquisition are due to them having slightly different coverage.

| Acquisition Parameters | | No. of Seedlings | Seedling Height Statistics (m) | | | | | Species | | |
|---|---|---|---|---|---|---|---|---|---|---|
| Phenology | GSD (cm) | | Min | Max | Mean | Median | StDev | %Sb | %Pj | %Lt |
| Leaf-on | 0.35 | 69 | 0.16 | 1.59 | 0.49 | 0.37 | 0.34 | 80 | 13 | 7 |
| | 0.75 | 176 | 0.15 | 2.38 | 0.59 | 0.43 | 0.44 | 82 | 6 | 13 |
| | 3 | 189 | 0.15 | 2.38 | 0.6 | 0.46 | 0.43 | 79 | 5 | 15 |
| Leaf-off | 0.35 | 69 | 0.15 | 1.59 | 0.49 | 0.37 | 0.33 | 78 | 13 | 9 |
| | 0.75 | 189 | 0.15 | 2.38 | 0.6 | 0.46 | 0.43 | 79 | 5 | 15 |
| | 3 | | | | – | | | | | |

As part of the larger BERA study that involved a manned photogrammetric flight (flight data not used in this paper), nine 60 cm plastic targets were placed as ground control points (GCPs) in and around each site, three along the corridor and six in the adjacent forest (three at each side), ensuring visibility from a low-flying aircraft (Figure 1, upper-right panel). A global navigation satellite system (GNSS) base station receiver (Hemisphere S320 Base) was set up near the sites. A GNSS rover (Hemisphere S320 Rover with XF3 Data Collector) was used to measure the location of each GCP and surveyed seedling; these measurements were later differentially corrected using the base station data. The average horizontal and vertical RMSE of the corrected point locations (both targets and seedlings) were 0.9 cm and 1 cm, respectively.

*2.3. Drone Data*

2.3.1. Acquisition

Three different drones were used in this study, one for each target spatial resolution: a DJI Mavic Pro, a DJI Inspire-2, and a Sensefly eBee (Table 2). The latter was not available in the leaf-off (October 2017) trial, so only the leaf-on (August 2017) trial has the three resolutions. There was some snow on the ground for the leaf-off trial, varying from a thin crust in more exposed upland locations to a foot or more deep in some protected lower pockets, which could potentially affect the detection of small seedlings. Preferred flying conditions were overcast and calm at midday (when the sun was at higher elevation), to minimize shadows. Cameras were generally aimed at nadir, and camera settings were generally left at defaults, such as auto exposure to adjust when lighting changes mid-flight.

The coarsest 3 cm GSD data were obtained using a Sensefly S.O.D.A. camera mounted on a fixed-wing eBee flying along predefined, parallel flight lines at 120 m above ground level (AGL) with 85% front overlap and 90% side overlap. The 0.75 cm GSD data were captured by a Zenmuse X4S gimbaled camera (model FC6510) mounted on an Inspire-2 flying at 30 m AGL (approximately 10 m above the forest canopy) along a predefined flight from one end of the 150 m-long site to the other, passing back and forth from side to side of the corridor in 5- to 10 m orthogonal segments and hovering at predefined capture points to achieve 90% side and front overlap. The 0.35 GSD photos were obtained only for one belt plot per site by manually piloting a Mavic Pro at 5 m AGL (inside the line, i.e., below the surrounding canopy) in DJI Go 4 Tripod mode, which imposes a top speed of 1.0 m/s for additional stability control, minimizing moving camera motion blur and also mitigating the risk of flying among protruding branches and other obstacles present in the seismic line at this height. A pattern of passes flying slowly up and down the belt plot and from side to side while triggering the built-in camera at 5 s intervals resulted in a reasonable photographic overlap, generally exceeding 80%.

**Table 2.** Drone acquisition and processing parameters. Mean AGL = mean flight altitude above ground level, SfM GCP err = ground control point positional error from the structure-from-motion workflow, GSD = output ground sampling distance. The density column refers to the output drone-based image point cloud (DIPC) dataset.

| Site | Acq. Date | Mean AGL (m) | Flight Conditions | Drone | Camera | Focal Len. (mm) | SfM GCP Err. (cm) | GSD (cm) | Approx. Density (pts/m²) |
|---|---|---|---|---|---|---|---|---|---|
| 460 | 3 August 2017 | 5 | Overcast; variable light winds | DJI Mavic Pro | FC220 | 4.73 | 21.5 | 0.35 | 25,000 |
| | | 31.5 | | DJI Inspire 2 | FC6510 | 8.8 | 20.9 | 0.75 | 7000 |
| | | 122 | Variable sun | eBee Sensefly | S.O.D.A | 10.2 | 3 | 3 | 650 |
| | 19 October 2017 | 5 | Sunny; gentle wind gusts | DJI Mavic Pro | FC220 | 4.73 | 22.3 | 0.35 | 32,000 |
| | | 31.7 | | DJI Inspire 2 | FC6510 | 8.8 | 12.1 | 0.75 | 5000 |
| 464 | 3 August 2017 | 4.7 | Sunny; scattered cloud | DJI Mavic Pro | FC220 | 4.73 | 25.9 | 0.35 | 30,000 |
| | 4 August 2017 | 30.4 | Occasional sun; increasing winds | DJI Inspire 2 | FC6510 | 8.8 | 27.2 | 0.75 | 5000 |
| | 3 August 2017 | 122 | Variable sun | eBee Sensefly | S.O.D.A | 10.2 | 5.7 | 3 | 940 |
| | 20 October 2017 | 4.5 | Sunny; gentle wind gusts | DJI Mavic Pro | FC220 | 4.73 | 11.3 | 0.35 | 30,000 |
| | | 30.2 | | DJI Inspire 2 | FC6510 | 8.8 | 78 | 0.75 | 4600 |
| 466 | 3 August 2017 | 5 | Occasional sun | DJI Mavic Pro | FC220 | 4.73 | 12.8 | 0.35 | 25,000 |
| | | 31 | | DJI Inspire 2 | FC6510 | 8.8 | 19 | 0.75 | 4000 |
| | | 122 | | eBee Sensefly | S.O.D.A | 10.2 | 5.6 | 3 | 735 |
| | 19 October 2017 | 5 | Sunny; gentle wind gusts | DJI Mavic Pro | FC220 | 4.73 | 8 | 0.35 | 38,000 |
| | | 30.5 | | DJI Inspire 2 | FC6510 | 8.8 | 9.2 | 0.75 | 3700 |

2.3.2. Drone-Based Image Point Cloud (DIPC) Generation

Agisoft PhotoScan 1.4.1 (build 5925; PhotoScan is now called Metashape) was used to process all drone imagery in a structure-from-motion workflow (SfM, a photogrammetric technique for estimating three-dimensional structures from two-dimensional image sequences). Images were aligned, GCP coordinates were imported, and GCP targets were identified in each corresponding photo. The images were bundle-adjusted in the selected coordinate system (EPSG 6655: UTM12 CSRS NAD83 + CVGD2013, https://epsg.io/6655) and a dense point cloud was built with the default values of "high quality" with "mild" depth filtering. Orthomosaics were created on the basis of the digital surface model derived from the dense point cloud and exported in TIFF format with a pixel size resampled to the nearest millimetre.

Because of the limited extent of the two finer GSD acquisitions, which only included a limited number of BERA GCP targets (see Section 2.2, last paragraph), there was a need for additional control points for co-registration and geo-referencing. For a given site, the coarsest GSD orthomosaic, which encompassed all nine BERA targets, was used to locate salient ground features identifiable in the finer GSD datasets (e.g., intersecting fallen logs). Once a sufficient number of these ground features were identified, coordinates were extracted and the ground elevation was taken from the weighed (by inverse distance) mean elevation of the four pixel centroids nearest to the feature in a digital terrain model (DTM) derived from airborne laser scanning (ALS) (see Section 2.4). The average predicted control point error resulting from the SfM workflow was 3 cm in the horizontal and 19 cm in the vertical. Predicted error was higher in datasets that used ground features as control points instead of BERA targets.

2.3.3. Softcopy Photogrammetry

We used the Stereo Analyst add-on module for ERDAS Imagine to compare the best height estimates derived from DIPC with manual estimates derived from softcopy photointerpretation of the same dataset. Belt plot 1 in site 464 had the most seedlings of all our sites, so we used it for this comparison. We used the leaf-off 0.35 cm GSD drone imagery (the dataset yielding the best DIPC results) for this analysis. Setting up the Stereo Analyst project required information on the focal length, flying height, and external orientation parameters for each photo in the set, which were exported from PhotoScan.

*2.4. Ancillary Data*

A 50 cm GSD bare-ground DTM derived from ALS was used as input to one of the methods determining ground elevation at the seedling locations, and also for estimating the elevation of the ground features used as additional control points (see Section 2.3.2). The ALS data were acquired by the Government of Alberta over the study area in August 2012 from a fixed-wing aircraft at 1500 m AGL. The ALS data was collected at a scan rate of 48.8 Hz, a pulse repetition rate of roughly 119,000 Hz, and a pulse footprint of 0.5 m. The acquisition achieved a density of over 2 pulses per $m^2$ with a projected vertical accuracy of 9 cm.

## 3. Methods

For each surveyed seedling in the study, we derived up to 15 different height estimates based on the available DIPC datasets. Each estimate comes from a different combination of GSD (0.35 cm, 0.75 cm, and 3 cm), phenological state (leaf-on and leaf-off), and ground determination method (based on either the DIPC itself or an ancillary digital terrain model). In addition, we estimated the height of a few seedlings using manual digital stereo photogrammetry (a.k.a. softcopy). Finally, we assessed the accuracy of the results using six accuracy metrics. Details follow.

*3.1. Height Estimation*

3.1.1. DIPC

For each seedling location and DIPC dataset, the height *h* of the seedling was estimated using the following equation:

$$h = z_{MAX} - z_{GROUND} \tag{1}$$

where $z_{MAX}$ is the elevation of the tallest DIPC point inside a vertical cylinder of 20 cm radius centered at the location, and $z_{GROUND}$ is the ground elevation at that location, estimated according to one of three methods described below. The 20 cm radius was found to be a good compromise between ensuring that the apex of the seedling is inside the cylinder, and avoiding the inclusion of taller points corresponding to adjacent shrubs. Any point 3 m taller than $z_{GROUND}$ was removed from the cylinder as it would have belonged to branches of tall neighboring trees. Several ground determination methods were tested, but after assessing preliminary results, we selected three methods for further analysis: (1) the elevation of the lowest point within a 1 m radius of the seedling location (hereafter the DIPC$_{MIN}$ method, which follows from Chen et al. [10]); (2) the weighed (by inverse distance) mean elevation in the 50 cm ALS DTM of the four pixel centroids nearest to the seedling location (hereafter the DTM$_{ALS}$ method); and (3) the weighed mean elevation of the three closest nodes in the DIPC triangular irregular network (TIN, hereafter the DIPC$_{TIN}$ method). The TIN was created from DIPC ground points, where the ground was classified using a method adapted from Isenburg [21] that isolates local minima and removes erroneous points depending on the minimum distance expected with no ground occlusion. Then, for each DIPC dataset and ground determination method, the predicted height of each seedling was compared with its observed (ground-measured) height (observed and predicted heights are available in Supplementary File S1). There were three GSD (0.35 cm, 0.75 cm, and 3 cm), two phenological states (leaf-on and leaf-off), and three ground determination methods (DIPC$_{MIN}$, DTM$_{ALS}$, and DIPC$_{TIN}$) under comparison, for a total of 15 different results (no leaf-on for the 3 cm GSD). To assess separately the two survey scenarios considered (survival and establishment), the seedling dataset was split into small ($\leq$30 cm) and regular (>30 cm) seedlings. The split threshold is double the height requirement for the survival survey (15 cm) and half that for the establishment survey (60 cm).

### 3.1.2. Softcopy Photogrammetry

We measured seedling heights using the 3D Measure tool in Stereo Analyst. The 3D Measure tool requires the user to judge the 3D position of the tip and base position of each seedling using the 3D floating cursor. Seedling heights are calculated by subtracting the Z field of each position. The work was performed by a trained softcopy analyst, who was given the task of measuring the height of all seedlings he could locate within belt plot 1 in site 464 using the aligned drone photos from the 0.35 cm GSD leaf-off acquisition.

The outcome of the softcopy analysis was a point shapefile with one attribute: height. Working then in ArcGIS, we identified the points that could be unambiguously matched to ground-measured reference seedlings as they appear in the 0.35 cm GSD leaf-off orthomosaic. A total of 22 such seedlings were identified out of 35 reference seedlings in the belt plot. We then compared the softcopy-measured heights with both the reference height and the estimated height from the best DIPC result.

### 3.2. Accuracy Assessment

For each of the 15 DIPC results (combination of GSD, phenology, and ground determination method), we compared the predicted height with the observed seedling height. To assess accuracy in each result, we derived six metrics (Equations (2)–(7)):

$$\text{RMSE} = \sqrt{\frac{1}{n} \sum_{i=1}^{n} \left(\widehat{h_i} - h_i\right)^2} \tag{2}$$

$$\text{MAE} = \frac{1}{n} \sum_{i=1}^{n} \left(\left|\widehat{h_i} - h_i\right|\right) \tag{3}$$

$$\text{MEDAE} = S\left(\frac{n}{2}\right); \ S = \left\{\left|\widehat{h_1} - h_1\right|, \left|\widehat{h_2} - h_2\right|, \ldots, \left|\widehat{h_n} - h_n\right|\right\} \mid \left|\widehat{h_i} - h_i\right| \leq \left|\widehat{h_{i+1}} - h_{i+1}\right| \tag{4}$$

$$\text{Bias} = \frac{1}{n} \sum_{i=1}^{n} \left(\widehat{h_i} - h_i\right) \tag{5}$$

$$R^2 = \left( \frac{n \sum h_i \widehat{h_i} - \sum h_i \sum \widehat{h_i}}{\sqrt{n \sum h_i^2 - (\sum h_i)^2} \sqrt{n \sum \widehat{h_i}^2 - \left(\sum \widehat{h_i}\right)^2}} \right)^2 \tag{6}$$

$$\overline{SD} = \frac{1}{m} \sum_{j=1}^{m} SD_{jk} \mid j \neq k; \; SD_{jk} = \frac{100}{n'} \sum_{i=1}^{n'} c_i \mid c_i = \begin{cases} 1 \; if \; \left|\widehat{h_{ik}} - h_{ik}\right| \leq \left|\widehat{h_{ij}} - h_{ij}\right| \\ 0 \; otherwise \end{cases} \tag{7}$$

where $h_i$ is the ground-measured height of seedling $i$, $\widehat{h_i}$ is its predicted height; $n$ is the number of seedlings covered by the DIPC dataset under assessment; $S(n/2)$ is the value of the middle element in the ordered list $S$ of absolute residuals (i.e., the median absolute error, MEDAE, not to be confused with the mean absolute error, MAE); $\overline{SD}$ is the mean stochastic dominance of the reported result over all other results; $m$ is the total number of results ($m = 15$); and $SD_{jk}$ is the stochastic dominance of result $k$ over result $j$, which we define as the percentage of $n'$ seedlings common to $j$ and $k$ where the absolute residual in $k$ is less than that in $j$. That is, a result with a $\overline{SD}$ value of 67% means that if we randomly pick a seedling and compare its height estimate from this result with that from a randomly selected result, and repeat this experiment many times, the result with $\overline{SD} = 67\%$ will yield a more accurate estimate than the other results 67 out of 100 times. MEDAE and SD are robust to outliers and can be tested for significance.

To test whether the MEDAE of the result with the lowest MEDAE is significantly smaller than the MEDAE of the other results, we ran the Stats R package wilcox.test function [22] with the argument $y$ equal to the absolute residuals of the best result; $x$ equal to the set of absolute residuals in each alternative result; paired = "false"; alternative = "greater"; and conf.level = 0.95. The significance of the stochastic dominance of the result with the highest $\overline{SD}$ over each of the other results was assessed by testing the null hypothesis that given a pair of absolute residuals $Rji$ and $Rki$ from a random seedling $i$ and results $j$ and $k$, $Rji$ and $Rki$ are equally likely to be greater than the other. To test this, we ran the BSDA R package SIGN.test function [23] with the argument $x$ equal to the set of absolute residuals in the result being compared; the argument $y$ equal to the absolute residuals of the best result in terms of $\overline{SD}$; md = 0; alternative = "greater"; and conf.level = 0.95. Finally, to assess how height errors would translate into counting errors in the survival and establishment surveys, we computed for each result errors of omission (i.e., not counting a seedling taller than the height requirement because its estimated height happened to be smaller than that) and errors of commission (i.e., counting a seedling shorter than the height requirement because its estimated height exceeded it).

To facilitate the discussion, a few additional assessments were made. First, to assess how seedling size affects the accuracy of height estimates (Breusch–Pagan tests indicated the residuals are heteroscedastic), we derived trend lines for RMSE, MEDAE, magnitude of bias, and $R^2$ for the best result at each GSD (so for three results). Since there are several accuracy metrics, what is "best" was determined as the mean combined ranking in those metrics. To avoid the derived trend being affected by outliers, we removed from this analysis seedlings with an absolute residual exceeding twice the RMSE of the corresponding result; such seedlings constituted less than 10% of the seedlings in any of the results. Then we ordered the list of seedlings of each of the three results in ascending order by reference height, selected seedlings 1 to 15 in each list, computed the value of these four metrics on the basis of those 15 seedlings only, and assigned as abscissa their mean reference height, thereby obtaining the first point in the trend graph. Then we repeated the same operation for seedlings 2–17, 4–19, 6–21, and so on until the end of the list was reached. The trend line of the resulting scatterplot was fitted using the Deming R package thielsen function [24], a robust regression method that also provides confidence intervals for the slope and intercept. Finally, to assess the vertical error of the three ground determination methods, RMSE and bias were also computed between the ground elevation estimated by each method and the observed GNSS elevation value for all seedling locations.

## 4. Results

### 4.1. Height Estimation

#### 4.1.1. DIPC, Seedlings ≤30 cm

Although the results for small seedlings yielded slightly better RMSE and MEDAE than those for taller (>30 cm) seedlings, their coefficients of determination $R^2$ indicate there is no relationship between predicted and observed heights for small seedlings ($R^2 < 0.1$ for all results). Only two out of 15 results have a %RMSE <100 (%RMSE relative to the mean observed height of seedlings in the result): the 0.75 cm leaf-off $DTM_{ALS}$ (RMSE = 16 cm, %RMSE = 73) and the 0.75 cm leaf-off $DIPC_{TIN}$ (RMSE = 19 cm; %RMSE = 85), but their $R^2$ is nil (<0.01) (Table S1). A possible explanation for the lack of relationship is that most of the small seedlings are surrounded by other low-lying vegetation that may create points of elevation similar to the apex of the seedling. Since the apex may sometimes not even generate a point, the estimated height hardly depends on the seedling itself. This may be more evident in results that used the $DTM_{ALS}$ method for ground determination, which include many seedlings for which the estimated height is 0, meaning that the tallest point in the local DIPC had an elevation equal to or lower than the $DTM_{ALS}$. For example, over half of the seedlings in the 0.75 cm leaf-on $DTM_{ALS}$ result have an estimated height of zero, yet this result has an RMSE of 33 cm and an underestimation bias of just 3 cm. This is an indication that RMSE and bias alone are not enough to assess the goodness of a result. An inspection of MEDAE and SD reveals that this particular result has the worst median absolute error (25 cm) and stochastic dominance (19%, meaning that this result performs worse than the other results in four out of five seedlings). Perhaps the clearest sign of the general lack of relationship is the fact that the top-ranking result is different for each of the accuracy metrics, something that could be expected if the results were ranked at random. For an example of the scatter between predicted and observed heights for seedlings ≤30 cm, see the red points in Figure 2, corresponding to the 0.35 cm leaf-off $DIPC_{TIN}$.

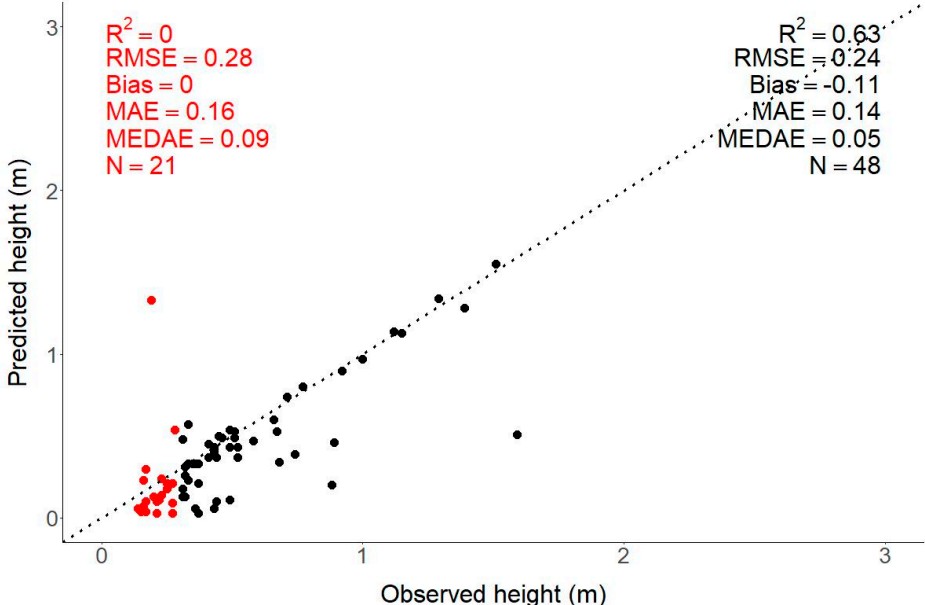

**Figure 2.** Scatterplot of observed (ground-measured) versus predicted heights in the best result for seedlings >30 cm (0.35 cm GSD, leaf-off, $DIPC_{TIN}$), including accuracy metrics (in black). The red dots represent seedlings ≤30 cm in this dataset, which were used only for the small-seedling analysis (corresponding accuracy metrics in red). MAE = mean absolute error; MEDAE = median absolute error.

### 4.1.2. DIPC, Seedlings >30 cm

The best results were obtained with a GSD of 0.35 cm, in leaf-off conditions using ground estimated from DIPC$_{TIN}$, which achieved an RMSE of 24 cm (%RMSE = 40%) and an $R^2$ of 0.63 (Figure 2; Table 3). The worst results came from the 3 cm GSD dataset, where $R^2$ was 0.13 or lower, and RMSE ranged from 67 cm (DTM$_{ALS}$ ground method) to 75 cm (DIPC$_{TIN}$) (Table 3). The 0.35 cm leaf-off DIPC$_{TIN}$ yielded the best values in all accuracy metrics except for bias and $R^2$ (the best coefficient of determination, $R^2$ = 0.67, was obtained with the exact same DIPC dataset but using the DIPC$_{MIN}$ method). Wilcoxon unpaired tests indicate that except for the 0.35 cm leaf-on DIPC$_{TIN}$, this result yields significantly lower MEDAE than the rest (indeed, it is remarkable that half of the seedlings have an absolute residual of 5 cm or less). On average, the 0.35 cm leaf-off DIPC$_{TIN}$ outperforms the other results in two-thirds of the seedlings ($\overline{SD}$ = 67%), although only half of the pairwise comparisons are statistically significant.

A graphical representation of bias versus RMSE reveals some general patterns (Figure 3). RMSE grows with GSD, but there is no clear trend with phenology or ground determination method. Eleven results out of 15 show underestimation bias, which tends to be larger in magnitude for leaf-off acquisitions. The ground determination method makes a difference, except for the leaf-on 0.35 cm GSD, for which the RMSE for the three methods is about the same.

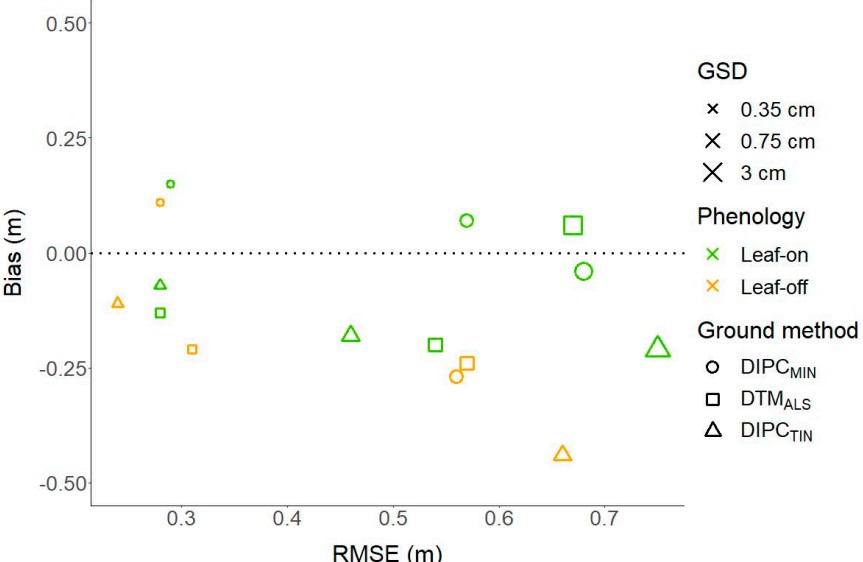

**Figure 3.** Seedling height estimation bias versus root mean square error (RMSE) for all ground sampling distance (GSD), phenology, and ground determination methods. DIPC$_{MIN}$ = ground elevation set to the minimum of the local DIPC; DTM$_{ALS}$ = ground elevation based on the light detection and ranging (LiDAR) digital terrain model; DIPC$_{TIN}$ = ground elevation based on a triangular irregular network derived from the DIPC.

**Table 3.** Results for seedlings >30 cm, ordered by the mean value of their ranking in each of the six metrics (e.g., the mean ranking for the 0.35 cm leaf-off DIPC$_{TIN}$ is (1 + 6 + 1 + 1 + 2 + 1)/6 = 2)). Note that the best value for each metric appears in bold. $N$ is the number of seedlings included in each result, $\bar{h}$ is the mean observed height of those seedlings in m, RMSE is the root mean square error, MAE is the mean absolute error, MEDAE is the median absolute error, R2 is the coefficient of determination, $\overline{\%SD}$ is the mean stochastic dominance of the result as defined in 3.2, and %om and %com are the percentage of omission and commission error respectively. All metrics given in m, except for R2 which is unitless, and $\overline{\%SD}$, %om and %com, which are given in percent. An asterisk in the MEDAE column indicates that according to the Wilcoxon test, the MEDAE of that result is significantly greater than the MEDAE of the first result in the table ($p < 0.05$). An asterisk in the $\overline{\%SD}$ column indicates that the sign test between that result and the first result was significant ($p < 0.05$), meaning that that first result yields a significantly larger proportion of seedlings where the absolute residual is less than that of the result with the asterisk.

| Rank | GSD (cm) | Phenology | Ground Method | $N$ | $\bar{h}$ (m) | RMSE (m) | BIAS (m) | MAE (m) | MEDAE (m) | $R^2$ | $\overline{\%SD}$ | %om | %com |
|---|---|---|---|---|---|---|---|---|---|---|---|---|---|
| 2.0 | 0.35 | leaf-off | DIPC$_{TIN}$ | 48 | 0.61 | **0.24** | −0.11 | **0.14** | **0.05** | 0.63 | **67** | 13 | 0 |
| 3.2 | 0.35 | leaf-on | DIPC$_{TIN}$ | 48 | 0.61 | 0.28 | −0.07 | 0.18 | 0.11 | 0.43 | 65 | 8 | 6 |
| 3.8 | 0.35 | leaf-off | DIPC$_{MIN}$ | 48 | 0.61 | 0.28 | 0.10 | 0.20 | 0.18 * | **0.67** | 57 | 4 | 19 |
| 5.3 | 0.35 | leaf-on | DTM$_{ALS}$ | 46 | 0.61 | 0.29 | −0.14 | 0.21 | 0.14 * | 0.51 | 52 | 7 | 0 |
| 5.8 | 0.75 | leaf-on | DIPC$_{MIN}$ | 123 | 0.75 | 0.57 | 0.06 | 0.33 | 0.13 * | 0.26 | 64 | 10 | 9 |
| 6.8 | 0.35 | leaf-on | DIPC$_{MIN}$ | 46 | 0.61 | 0.29 | 0.15 | 0.24 | 0.22 * | 0.52 | 47 * | 4 | 28 |
| 7.3 | 0.35 | leaf-off | DTM$_{ALS}$ | 48 | 0.61 | 0.32 | −0.22 | 0.24 | 0.20 * | 0.59 | 44 * | 15 | 0 |
| 7.3 | 3.00 | leaf-on | DTM$_{ALS}$ | 135 | 0.75 | 0.67 | 0.06 | 0.39 | 0.18 * | 0.13 | 61 | 12 | 7 |
| 8.8 | 0.75 | leaf-on | DIPC$_{TIN}$ | 123 | 0.75 | 0.47 | −0.19 | 0.34 | 0.24 * | 0.37 | 39 * | 22 | 5 |
| 9.8 | 0.75 | leaf-off | DTM$_{ALS}$ | 135 | 0.75 | 0.57 | −0.25 | 0.38 | 0.19 * | 0.07 | 58 | 30 | 2 |
| 10.3 | 3.00 | leaf-on | DIPC$_{MIN}$ | 135 | 0.75 | 0.68 | **−0.04** | 0.45 | 0.28 * | 0.10 | 45 * | 19 | 7 |
| 10.5 | 0.75 | leaf-off | DIPC$_{MIN}$ | 135 | 0.75 | 0.57 | −0.28 | 0.40 | 0.27 * | 0.12 | 54 | 33 | 1 |
| 10.8 | 0.75 | leaf-on | DTM$_{ALS}$ | 123 | 0.75 | 0.54 | −0.20 | 0.41 | 0.34 * | 0.22 | 35 * | 20 | 16 |
| 13.8 | 0.75 | leaf-off | DIPC$_{TIN}$ | 135 | 0.75 | 0.67 | −0.45 | 0.54 | 0.39 * | 0.10 | 22 * | 40 | 0 |
| 14.2 | 3.00 | leaf-on | DIPC$_{TIN}$ | 135 | 0.75 | 0.75 | −0.22 | 0.57 | 0.41 * | 0.09 | 23 * | 33 | 6 |

### 4.1.3. Softcopy Photogrammetry

The softcopy estimates of seedling height using the best dataset (0.35 cm GSD leaf-off) yielded the same RMSE as their DIPC counterparts (Figure 4). The median absolute difference between softcopy and DIPC estimates is 9 cm. A paired *t*-test between the two sets of height estimates revealed no significant difference ($p = 0.20$).

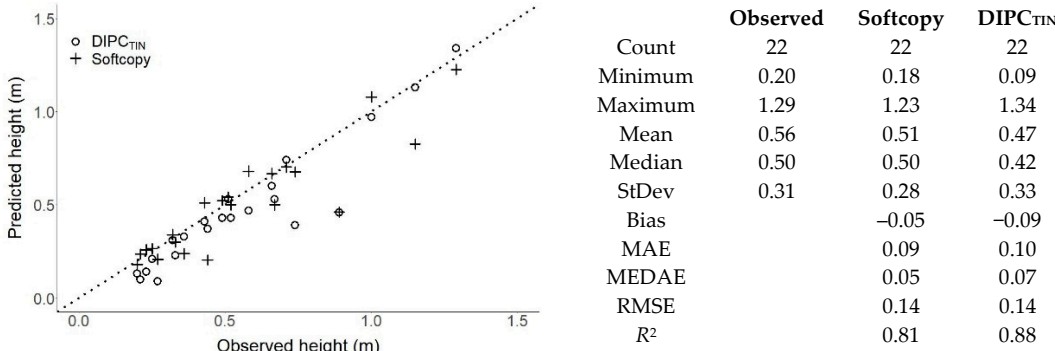

| | Observed | Softcopy | DIPC$_{TIN}$ |
|---|---|---|---|
| Count | 22 | 22 | 22 |
| Minimum | 0.20 | 0.18 | 0.09 |
| Maximum | 1.29 | 1.23 | 1.34 |
| Mean | 0.56 | 0.51 | 0.47 |
| Median | 0.50 | 0.50 | 0.42 |
| StDev | 0.31 | 0.28 | 0.33 |
| Bias | | −0.05 | −0.09 |
| MAE | | 0.09 | 0.10 |
| MEDAE | | 0.05 | 0.07 |
| RMSE | | 0.14 | 0.14 |
| $R^2$ | | 0.81 | 0.88 |

**Figure 4.** Scatterplot of observed (ground-measured) versus predicted heights for seedlings measured both with softcopy (crosses) and drone-based image point clouds (DIPC) (circles), including descriptive stats and accuracy metrics. DIPC$_{TIN}$ = ground elevation estimated the weighed mean elevation of the three closest nodes in the DIPC triangular irregular network; RMSE = root mean standard error; MAE = mean absolute error; MEDAE = median absolute error.

### 4.2. Counting Errors in Seedling Surveys

Height-estimation errors can lead to counting errors in automated seedling surveys if the estimated height of a detected seedling is lower than the required height but the true height is above it (that would be an omission error), or vice versa (commission error). For the small seedlings (survival survey scenario), the counting errors that each result would produce were highly variable, from 0% omission (0.35 cm leaf-on DIPC$_{MIN}$, which had the largest overestimation bias, 31 cm) to 93% (0.75 cm leaf-off DIPC$_{TIN}$, which had the most severe underestimation bias, 17 cm). Since no seedling was shorter than the 15 cm requirement, commission errors could not be assessed for the small seedling set.

For seedlings >30 cm (establishment survey scenario; height requirement 60 cm), all results but two produced more omission than commission errors (Table 3). Omission rates ranged from 4% to 40%, while commission rates ranged 0% to 28%. The best result in terms of height accuracy (0.35 cm leaf-off DIPC$_{TIN}$) did not result in the lowest counting errors, but the error rates were reasonable (13% omission and no commission). The best compromise would come from the 0.35 cm leaf-on DIPC$_{TIN}$ (second best in height accuracy), which had an 8% omission rate and a 6% commission rate. Interestingly, there are results at coarser resolutions that would still provide a decent count assuming perfect detection, such as the 0.75 cm leaf-on DIPC$_{MIN}$ (10% omission rate and 9% commission rate), and the 3 cm leaf-on DTM$_{ALS}$ (12% omission rate and a 7% commission rate). This suggests that height estimation errors would only have a small impact on seedling counting errors in establishment surveys, probably smaller than detection errors.

## 5. Discussion

Several factors influenced our results: GSD, phenology, ground determination method, seedling size, and measurement errors. We discuss them separately, highlight the contributions and limitations of our study, and provide an outlook for seedlings surveys based on this technology.

### 5.1. Effect of Ground Sampling Distance (GSD)

GSD is more influential than phenology or ground determination method for estimating conifer seedling height using DIPC. The four best results in the ranking of Table 3 include all phenologies and ground determination methods, but only one GSD: the finest (0.35 cm). Figure 3 also provides a clear pattern of increasing RMSE as GSD coarsens. Even if the finest GSD provided sufficient 3D reconstruction detail of seedlings from which to reliably estimate height, the results are unreliable for small seedlings (≤30 cm), probably because of the presence of points from adjacent low vegetation within the 40 cm diameter cylinders used to clip the DIPC. One drawback of the finer GSD, however, is the presence of below-ground noise (spurious points, shown in red in Figure 5). The overestimation bias found with the $DIPC_{MIN}$ ground determination method is probably caused by this noise. Wind may also have played a role. There were gentle wind gusts during some of the acquisitions and moderate turbulence was created by the Mavic Pro when it flew at 5 m AGL to achieve a GSD of 0.35 cm. As Frey et al. [25] explain well, a finer GSD exacerbates wind effects on smaller scene elements that could cause a shift of location in adjacent photos taken even a second apart. Therefore, the finer the GSD, the more influence wind will have. However, given that the best results were nonetheless achieved with the finest GSD, we can assume that, at least for our acquisitions, the detrimental effect of wind did not offset the benefits of a finer GSD. A coarser GSD is less affected by wind, but the flattening effect that can be appreciated in Figure 5 (also in Video S1) leads to a less reliable estimation and a weaker correlation. This finding is consistent with the observation of Frey et al. [25] that a finer GSD and high image overlap are both beneficial for sampling lower parts of the canopy, understory, and ground. In short, in our study, seedling height estimation accuracy and correlation were stronger at finer GSDs; the RMSE at 0.35 cm GSD was less than half that at 3 cm GSD, and their highest $R^2$ values were 0.67 and 0.13, respectively.

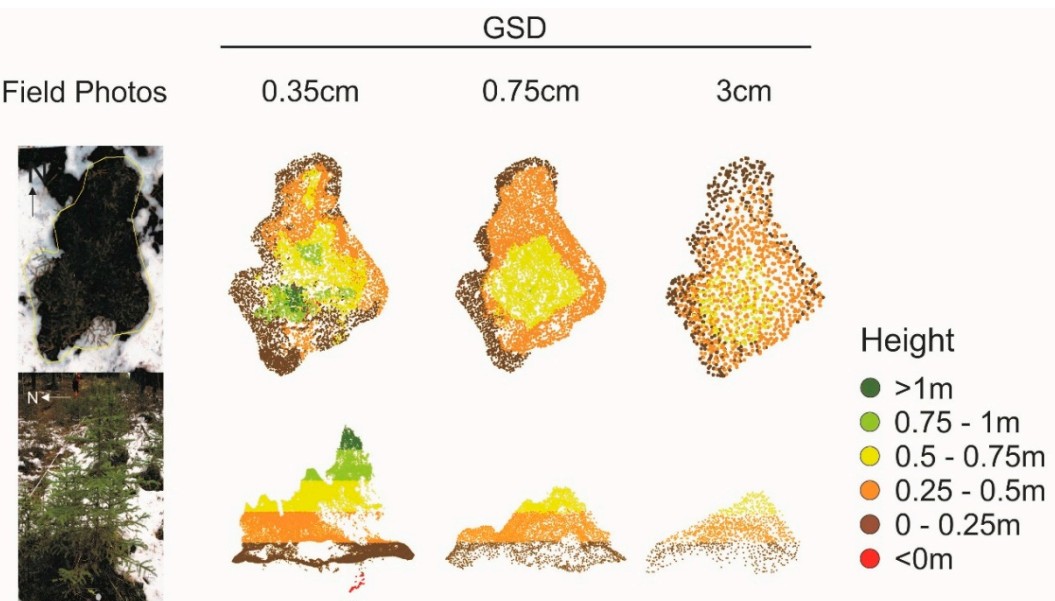

**Figure 5.** An example illustrating relative point densities, 3D reconstruction detail, and height estimation for a cluster of spruce seedlings (drone and ground photos on the left) using each of the three ground sampling distances (GSD) studied. Points colored according to their height. Note that as GSD coarsens, the corresponding point cloud "flattens" (see also Video S1).

### 5.2. Effect of Leaf Phenology

While the best result was obtained in leaf-off conditions, its MEDAE and SD were not significantly better than those of the second-best result, which was leaf-on (Table 3). A *t*-test between the height estimates of the best and second-best results revealed no significant statistical difference ($p = 0.18$),

and we cannot appreciate major differences in the example of Figure 6 except for the tallest seedling. Imangholiloo et al. [15] also found that for plots with taller young trees, height prediction was negligibly different between leaf-on and leaf-off conditions. Their study and ours suggest that phenology does not play a crucial role when estimating height for more mature conifer seedlings, which is relevant for establishment surveys. However, for small seedlings, Imangholiloo et al. [15] obtained a relative RMSE of 26.4% and 9.6% (30 cm and 11 cm, respectively) for mean height of spruce seedlings in leaf-off and leaf-on conditions, respectively. It is unclear why the leaf-on performed so much better than the leaf-off for their five young spruce plots. They mention the poor reconstruction of leafless branches, which in our case did play a role for larch (Figure 8b), but that does not apply to spruce. A possible explanation is that their object-based classification of the leaf-on orthomosaic created larger seedling image-objects than in the leaf-off, thus they tended to encompass more adjacent small seedlings. As a result, these small trees were excluded from the computation of the plot's mean height, hence reducing the inherent DIPC underestimation bias as well as the RMSE.

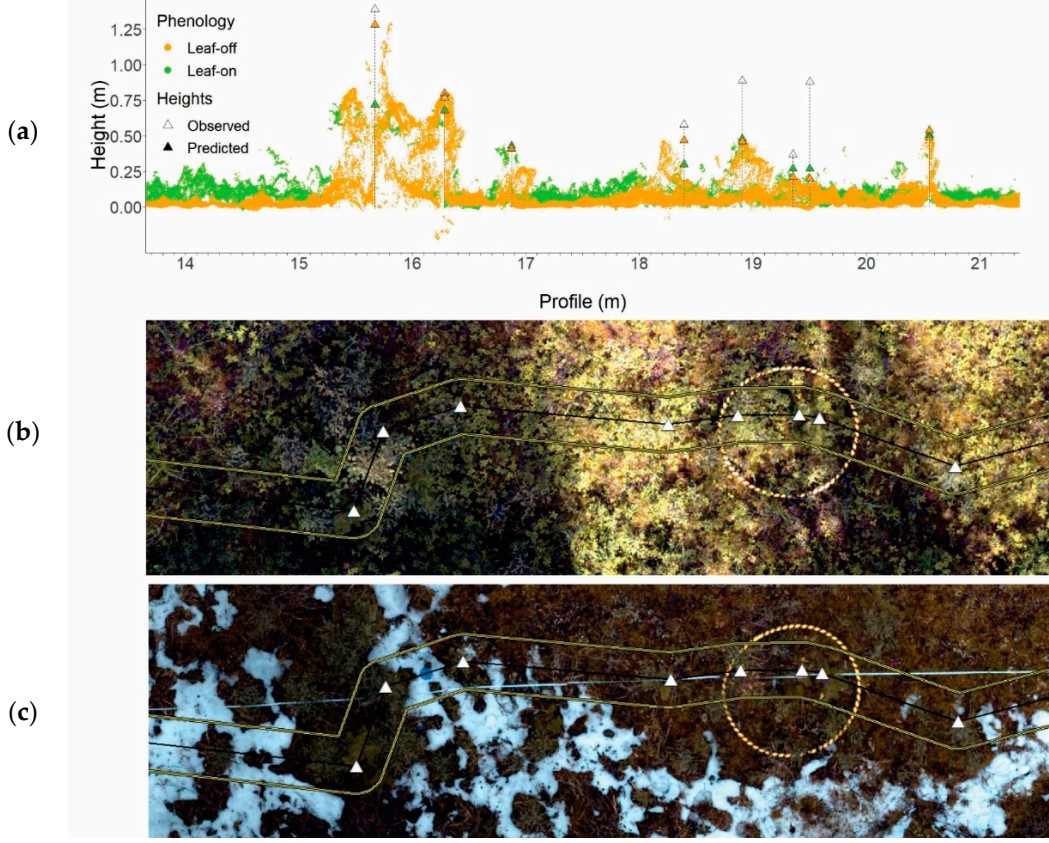

**Figure 6.** A sample vertical profile from the 0.35 cm ground sampling distance (GSD) drone-based image point clouds (DIPCs) (**a**), and respective leaf-on (**b**) and leaf-off (**c**) orthomosaics. The profile was extracted from a 20 cm wide transect (black line with yellow buffer) connecting several surveyed seedlings (white triangles). The height estimates come from the DIPC$_{TIN}$ method (the weighed mean elevation of the three closest nodes in the DIPC triangular irregular network). The scale of the vertical profile is slightly coarser than that of the displayed images, because the transect is not straight. Ground photos of the 1-m diameter, red and yellow hula-hoop, and a 3D view of the corresponding point clouds from different GSD can be watched in Video S1.

In our case, underestimation is typically more pronounced in leaf-off conditions (i.e., compare the relative position along the y axis of the green and orange icons of the same shape and size in Figure 3). In leaf-on conditions, however, bias magnitude and MAE are farther apart than in leaf-off conditions for the DIPC-based ground determination methods, suggesting that there are more seedlings being

overestimated in leaf-on than in leaf-off conditions, which could be due to the presence of points generated by taller adjacent deciduous vegetation. Despite this, and given that the differences are not significant, we concur with Imangholiloo et al. [15] that it is preferable to do the acquisitions during the leaf-on period. Leaf-on has the additional advantage of being more predictable than the leaf-off period, which in the boreal is too cold to operate a drone except for the shoulder seasons.

*5.3. Effect of Ground Determination Method*

At the finest GSD, ground estimated from $DIPC_{TIN}$ consistently provides smaller absolute residuals than the other ground determination methods. This could be because $DIPC_{TIN}$ is not affected by below-ground noise from the data, whereas $DIPC_{MIN}$ is. Although $DIPC_{TIN}$ yielded the best prediction of seedling height, it does not follow that it provides the best estimate of ground elevation. Comparison of elevation values obtained from each ground determination method with GNSS elevation values revealed that $DTM_{ALS}$ consistently followed the GNSS-measured elevation better than $DIPC_{TIN}$, with a bias of −2 cm and an RMSE of 20 cm for the 69 seedling locations in the 0.35 cm leaf-off dataset, whereas the $DIPC_{TIN}$ had an RMSE of 25 cm and a bias of −13 cm (for comparison, these figures are 47 cm and −36 cm, respectively, for the $DPIC_{MIN}$). Given the poorer height estimation performance using $DTM_{ALS}$, it is likely that the point clouds carry local spurious deformations of the true terrain shape that make an ancillary ALS DTM unsuitable for estimation of the height of low vegetation such as seedlings. This is consistent with previous studies, such as Salach et al. [26], who found that the SfM workflow leads to a point cloud that represents terrain elevation with less fidelity than an ALS point cloud. Nevertheless, given that height estimation is relative and given that the top point in the seedling cylinders carries the same local elevation error as the ground points, $DIPC_{TIN}$ outperforms $DTM_{ALS}$. This means that an ancillary ALS DTM is not required, as has been pointed out by the authors of earlier studies [10,27]. The $DIPC_{TIN}$ method, however, is more sensitive to ground occlusion and snow presence, and the production of a suitable TIN requires some user interaction. The $DIPC_{MIN}$ method, on the other hand, is simpler, but it is more sensitive to below-ground noise and slope. For coarser GSDs, the ground determination method appears to still make a difference, but the best method is different: $DIPC_{MIN}$ for 0.75 cm GSD (perhaps because of the absence of below-ground noise in this point cloud), and $DTM_{ALS}$ for 3 cm GSD (perhaps because of a better representation of the terrain of this point cloud, which had the largest extent and included the highest number of GCPs). Although it requires more user interaction, $DIPC_{TIN}$ is probably a safer alternative than $DIPC_{MIN}$, but more research is needed to confirm this.

*5.4. Effect of Seedling Size*

The trend analysis revealed that RMSE is expected to grow as the mean height of the surveyed seedlings increases for all three GSD analyzed (0.35 cm, 0.75 cm, and 3 cm). The rate of RMSE growth, however, is low: between 1.1 cm and 2.4 cm per 10 cm of increase in mean height for the 0.35 cm GSD (Figure 7), and 2 to 4 cm for the other GSDs (not shown). This means that the relative RMSE will decrease almost as fast as if the residuals were homoscedastic. However, this trend cannot be reliably extrapolated beyond the analyzed range of mean height (0.3 to 1 m for the 0.35 cm GSD, and 0.3 to 1.5 m for the other GSDs).

We found similar trends for the magnitude of bias, which increases between 0.9 cm and 1.8 cm per 10 cm of increase in mean height for the 0.35 cm GSD (Figure 7) and between 0.8 cm and 1.9 cm for the other GSDs (not shown). The MEDAE behaviour was different for the 0.35 cm GSD, which according to the derived trends could show homoscedasticity (no change) or even decrease with mean height (Figure 7), whereas in the other GSDs it would increase between 0.9 cm and 2.7 cm per 10 cm of increase in mean height. Finally, the trends in $R^2$ are less reliable given the wide confidence intervals and the absence of a clearly linear pattern, but there seems to be an upward trend at least for the 0.35 cm GSD (Figure 7). These results suggest that DIPC estimates should be more reliable for taller seedlings.

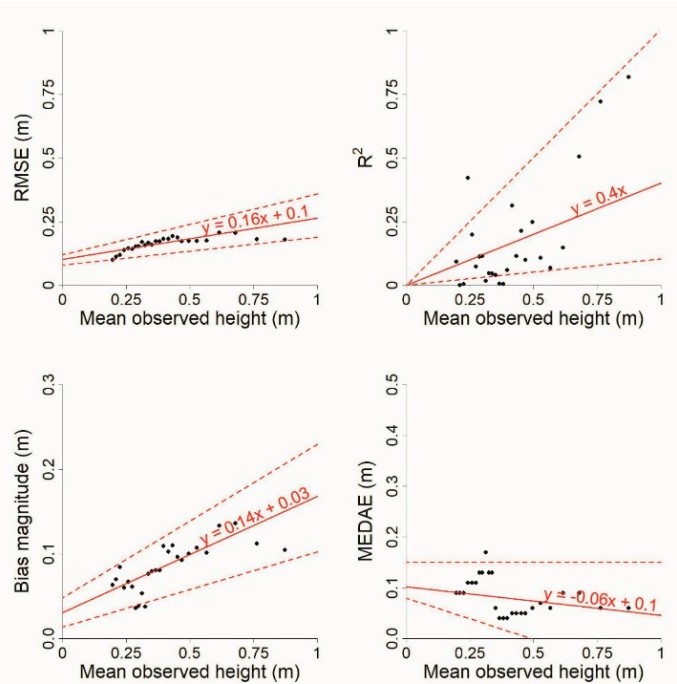

**Figure 7.** Trends in RMSE, magnitude of bias, median absolute error (MEDAE), and $R^2$ as a function of mean seedling height for the best result (0.35 cm GSD, leaf-off, $DIPC_{TIN}$). Each point represents a subset of 15 seedlings ordered in ascending order by reference height. Each pair of consecutive points along the x axis shares 13 seedlings. The solid red line is the Theil–Sen regression line, and the dashed lines represent the lower and upper 95% confidence intervals.

### 5.5. Errors Contributing to Outliers

The presence of outliers in our analysis suggests that our results are conservative, that is, the actual error that can be expected from an operational drone survey may be less than what we report, and it could be more in line with what we found for the 22 seedlings in the softcopy comparison (14 cm RMSE and 0.88 $R^2$, Figure 4). In particular, we found several sources of outlier-causing errors, some of which could be avoided or reduced in future studies:

1. *Location error*—all seedlings locations were visually checked to determine which of them did not align with their apparent position in the orthomosaics. Seventeen of the 189 seedlings in this study had an offset greater than 20 cm in at least one orthomosaic. There were several instances where the seedling could not be found at their measured location within any of the orthomosaics. However, most of those seedlings were obstructed from view by adjacent trees so we could not assess their location error. Location errors greater than 20 cm are likely to cause an outlier in the height estimation, which only considers DIPC points within a 20 cm radius of the reported locations. Location error is expected through the DIPC workflow, where XYZ coordinates can be shifted from their true geographic location because of complexities within the SfM process [28,29]. This was demonstrated by slight shifts found between orthomosaics of the different DIPC datasets, as well as between the orthomosaics and their input GCPs. Fortunately, in an operational drone survey, seedling locations will come directly from semi-automated seedling detection on the orthomosaic [5], so these locational offsets will be absent.

2. *Field measurement or data entry error*—seedlings that corresponded to obvious outliers in the 0.35 cm GSD leaf-off dataset were checked in relevant field photos or videos. Figure 8a shows an example of such an error where "observed" height is 0.19 cm, but orthomosaics and field photos show this value is probably incorrect. This type of errors inflates the observed RMSE, but we decided to include them because removing outliers is hardly justifiable except for the limited purposes of a sensitivity analysis such as the one we did on seedling size (Section 5.1).

3.  *3D reconstruction error*—for the finest GSD there was below-ground noise (discussed in Section 5.2). Whatever the source of below-ground noise is, it probably also creates spurious points elsewhere in the DIPC, which will affect height estimation if they happen to be local maxima. Additionally, seedlings can be occluded by adjacent taller vegetation, preventing a full reconstruction in the point cloud. As described earlier, wind will also have an impact on 3D reconstruction where the DIPC workflow will have trouble matching the image pixels of swaying objects. Species phenology and morphology also affect 3D reconstruction; bare twigs from deciduous larches will be difficult to fully capture (e.g., Figure 8b). It also seems that jack pine seedlings are harder to reconstruct, perhaps because of their narrow shape. All these effects lead to underestimation bias and are unlikely to be reduced in an operational drone survey, except perhaps by limiting the maximum ambient wind allowed for acquisitions, as suggested by Frey et al. [25].

4.  *Non-seedling point errors*—in a naturally regenerating linear disturbance there will be a lot of vegetation present that interferes with height estimation of nearby seedlings. There were several instances where a seedling's height was overestimated because of neighboring tall shrubs or adjacent mature trees. To reduce this effect, we limited the DIPC extraction to a narrow vertical cylinder design, which is simple, but the inclusion of some points from neighboring vegetation inevitably occurred. A procedure that included point cloud segmentation [30], wherein clusters of points belonging to the same plant are given a unique ID—thus allowing for the removal of neighboring points belonging to an adjacent plant—would solve this problem, although tuning such an algorithm for short vegetation is not trivial.

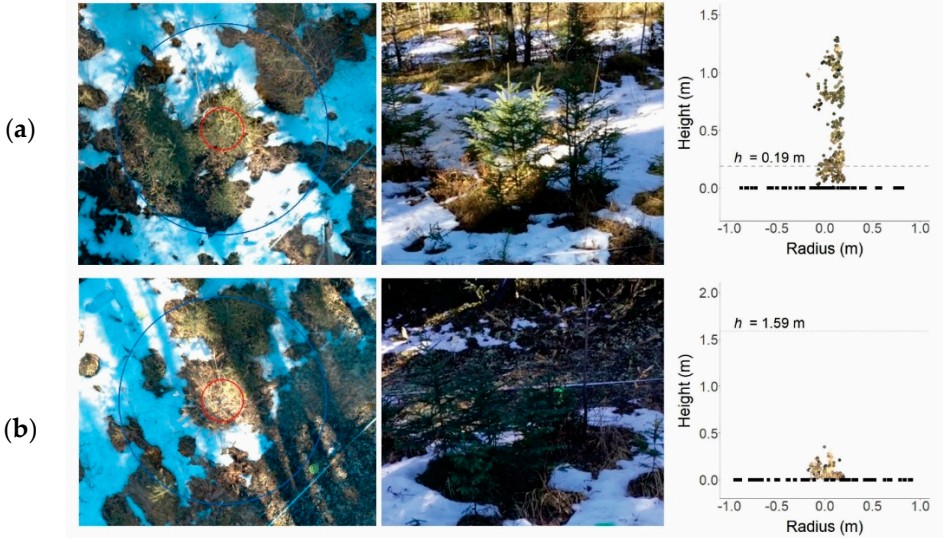

**Figure 8.** Example of (**a**) height overestimation of a spruce seedling (probably due to a data entry blunder) and (**b**) height underestimation of a larch seedling (leaf-off). The left panels are a nadir view of seedlings from the orthomosaic, the middle is a profile from a field photo, and the right is the 3D reconstruction.

*5.6. Study Contributions and Limitations*

To the best of our knowledge, our study is the most comprehensive analysis to date of the accuracy of height estimates for individual (rather than plot level) conifer seedlings derived from drone-based image point clouds (DIPC). We provide insights on the effects of GSD, phenology, ground determination method, seedling size, and other factors contributing to outliers. We include two accuracy metrics that other studies did not use, namely the median absolute error (MEDAE) and the mean stochastic dominance ($\overline{SD}$), which we define as the expected proportion of instances (seedlings) where a given method/result outperforms others. An advantage of these two metrics is that they can be statistically tested for significance, MEDAE through a Wilcoxon test, and SD through a sign

test. In addition, we went beyond the typical diagnostic for heteroscedasticity and were able to quantify trends in accuracy as a function of seeding height. Our study also used a finer GSD for conifer seedling height estimation than any of the previous studies, which provides better reconstruction of the target objects and probably accounts for the lower RMSE we achieved at the individual seedling level (Imangholiloo et al. [15] obtained a similar RMSE, but it referred to plot-level estimates). Ours is also the first DIPC study to include a separate analysis for conifer seedlings less than 30 cm in height. Finally, our study is the first to assess the implications of height estimation errors on counting errors for automated stocking surveys where seedlings must exceed a given height to be counted.

This study was conducted opportunistically in the context of the larger BERA project. While the three chosen sites portray different levels of regeneration, they are just a small subset of the sites surveyed by BERA. In particular, none of the three sites include actively restored seismic lines, where the most common treatment is mounding followed by the planting of seedlings on the top edge of the mounds. We anticipate mound microtopography will hamper seedling height estimation, so our results may not be extrapolated to treated sites. Our conclusions are based on a small sample size: 135 seedlings for the 3 cm GSD, approximately 120 for the 0.75 cm GSD, nearly 50 for the 0.35 cm GSD, and just 22 for the softcopy estimation. Although small, these samples can be considered random given the sampling design, so our conclusions should be free of any researcher bias, but future studies should expand the number of sites, tally all seedlings in the sites, and ensure the same coverage of seedlings for all drone datasets. In addition, each combination of GSD and phenology should have several replicate acquisitions to assess the effects of other factors not taken into consideration by this study such as wind, snow presence, and illumination, as well as camera orientation and model, image format, exposure, and SfM parameters.

*5.7. Outlook*

Given our results, we believe that the height of conifer seedlings taller than 30 cm can be reliably estimated using DIPC datasets with a GSD finer than 0.5 cm, with better accuracy than the typical ALS dataset. For example, in a study in Norway, Naesset and Nelson [31] measured the location and height of 342 small trees of Norway spruce, Scots pine (*Pinus sylvestris* L.), and downy birch (*Betula pubescens* Ehrh.), and compared these measurements with height estimates derived from 7.7 pulses/m$^2$ ALS data. The lowest RMSE was 41 cm (−40 cm bias and 11 cm standard deviation) for scots pine in the 0 to 1 m height class ($n = 29$), which is almost double what we obtained for our best result. Furthermore, the reliability of DIPC height estimates is expected to increase with seedling size. If the trends we found for RMSE could be extrapolated beyond the studied range, a relative RMSE of 10% could be achieved for mean seedling heights greater than 2 m in a scenario where ground points in the point cloud are available and correctly classified, and where interfering points from nearby tall vegetation are removed via segmentation. A GSD of 0.5 cm can be achieved with commercially available cameras mounted on drones flying over the forest canopy. It does not seem feasible using a manned aircraft, however. In addition to the GSD requirement, these aircraft could not easily achieve the large image overlap required for SfM.

Coupled with automated detection such as those in [5], a similar workflow could offer a cost-effective alternative to ground-based seedling surveys, which in addition to cost considerations would have the advantage of avoiding trampling over sensitive restored ground. For example, a trained artificial intelligence (AI) could detect the location of the seedlings, and other modules could make automated calls on species, health status, and height. Repeated surveys could add information on mortality and growth. Once beyond visual line of sight (BVLOS) drone flights are allowed in Canada, larger areas could be surveyed, providing economies of scale to a surveying company [32]. We hope that further studies will corroborate these expectations.

## 6. Conclusions

Our results suggest that drone-based image point clouds (DIPCs) can be used for estimating the height of individual conifer seedlings, provided GSD is fine enough to reconstruct the seedling apex. Our results also indicate that seedlings need to reach a certain size before height can be dependably estimated. Specifically, it is unlikely that drone surveys can be used for survival assessments (2–5 years after treatment) where seedlings are still small. Even at the finest GSD, the best relative RMSE we could obtain for seedlings ≤30 cm was 73%, and the best $R^2$ was 0.13. In contrast, for seedlings >30 cm, these figures greatly improved: they went to 40% (24 cm) and 0.67, respectively.

The limited comparison with height estimates derived from manual softcopy photogrammetry suggests that there are no significant differences with the automated DIPC height estimation at the finest GSD and, therefore, the automated procedure could replace the manual one with similar accuracy (RMSE of 14 cm for both methods in the subset of 22 matched seedlings). In terms of the impact of height estimation error on counting errors in drone-based, automated seedling surveys, it seems low regardless of the GSD (8% omission and 6% commission error rates for the result showing the lowest impact at 0.35 GSD, and 12% and 7%, respectively, for the 3 cm GSD). Hence the accuracy of seedling detection will be a more crucial factor in this type of surveys.

Seedling height estimation accuracy and correlation were stronger at finer GSDs, as expected. Half of the seedlings in the best result (0.35 cm GSD leaf-off DIPC$_{TIN}$) had an error of 5 cm or less, and this result outperformed the others in two-thirds of the seedlings. While the best result was leaf-off, it was not significantly better than the second-best result (leaf-on), so it is preferable to plan the acquisitions during the more predictable leaf-on period. Seedling heights estimated with ground elevation based on the DIPC itself were more accurate than heights estimated using an ancillary ALS-based DTM. While RMSE is expected to increase slightly with the mean height of seedlings, it does so at a slow rate, meaning that both %RMSE and $R^2$ are expected to improve with seedling size, at least at the finer GSD.

Several factors led to outliers that could be mitigated or prevented in an operational survey: mismatches between the seedling's reported location and its position in the orthomosaics, which would not occur when the seedling is directly detected on the orthomosaic; and spurious below-ground points and points belonging to adjacent vegetation, which could be removed by filtering and segmentation, respectively. Coupled with automated seedling detection routines, DIPC could produce a low-cost inventory of conifer seedlings growing in disturbed sites to monitor recovery in the boreal forest.

**Supplementary Materials:** The following are available online at http://www.mdpi.com/1999-4907/11/9/924/s1: Supplementary File S1: seedling height data, observed and estimated; Table S1: Results for seedlings ≤ 30 cm; Video S1: Sample visualization of several DIPCs for a circular 1-m² plot with seedlings.

**Author Contributions:** Conceptualization, G.C. and G.J.M.; methodology, G.C., M.F. and M.G.; validation, M.F., G.C. and G.J.M.; formal analysis, M.F.; investigation, G.C., M.F. and G.J.M.; resources, G.C. and G.J.M.; data curation, M.F. and M.G.; writing—original draft preparation, review, and editing, G.C., M.F., G.J.M. and M.G.; visualization, M.F., M.G.; supervision, G.C.; project administration, G.C.; funding acquisition, G.C. and G.J.M. All authors have read and agreed to the published version of the manuscript.

**Funding:** This research was supported by funding from Natural Resources Canada (NRCan) Office for Energy Research and Development (OERD), and a Natural Sciences and Engineering Research Council of Canada (NSERC) Collaborative Research and Development Grant (CRDPJ 469943-14) in conjunction with Alberta-Pacific Forest Industries, Cenovus Energy, ConocoPhillips Canada, and Canadian Natural Resources.

**Acknowledgments:** Man Fai Wu surveyed the seedlings and deployed the ground control points. Jennifer Hird performed the point post processing of seedling coordinates. William Gartrell did the Agisoft processing of the drone photos. Julia Linke designed the sampling design for the BERA study and provided coordination between the Canadian Forest Service and the University of Calgary teams. Francisco Mauro and two anonymous reviewers provided helpful comments to an earlier version of this manuscript. Jennifer Thomas did an editorial review of the final draft.

**Conflicts of Interest:** The authors declare no conflict of interest. The funders had no role in the design of the study; in the collection, analyses, or interpretation of data; in the writing of the manuscript; or in the decision to publish the results.

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
