# Peer review of "Estimating Individual Conifer Seedling Height Using Drone-Based Image Point Clouds"

_forests, doi:10.3390/f11090924_

Round 1

Reviewer 1 Report

Dear authors,

thank you for this study. Although the degree of novelty is average, you provide a robust assessment of UAV-based DIPC approaches to quantify conifer seedlings regeneration success, in terms of height and counting estimations, which in my opinion is timely and of high relevance in an operational context.

Overall, the study is presented well and you account for existing literature on the topic. I do have however a few suggestions and open points that I encourage you to address:

  1. UAV flight altitude: I think it is only mentioned in Table 2. However in a forestry context min altitude might be a limiting factor if regeneration rates need to be assessed close by / in-between mature forest stands. Also since different UAV-camera systems were employed in this study, the authors could perhaps provide recommendations as well on hardware configurations for such application.
  2. Effect of phenology: at certain times of the year (autumn/winter) the effect of background vegetation, and thus seedlings detection errors, might be reduced. Did the authors observe such an effect?
  3. From a more physiological point of view, I would expect that there is an optimal time in the season, coinciding with the phases following budburst, that would be best suited to determine whether seedlings are thriving. The assessment of regeneration success could be expanded beyond counting and height, to include also physiological traits linked with stress responses that can be assessed using vegetation indices based on specific wavelengths (see D'Odorico, P., et al. (2020), doi:10.1111/nph.16488). This is briefly mentioned in the Outlook, but in my opinion could be discussed a bit more in detail to give a more comprehensive picture of the potential of UAV assessments in the context of forest regeneration.
  4. The data pool used to validate estimates from UAV DIPC, and the validation approach in general, is not well presented. I understand that there is a two-stage validation, one based on the Stereo Analyst add-on module for ERDAS Imagine (called Softcopy Photogrammetry) and one based on ground measurements along the transect (+tallest in each plot). However, I find it confusing to understand when which validation is used and what is presented in the different figures.
  5. As a follow-up to my previous comment, I find Figure captions not containing sufficient information to understand the figure without having to consult the text. Please add details, for example Fig.2 how was Observed height obtained here).
  6. The manuscript is in my opinion a bit wordy, I would suggest shortening it where possible. For example information provided in Table 2 is partly repeated in the text. Further, I suggest merging Outlook and Conclusion sections. Although Conclusions need to be specific to the study, right now they do repeat part of what is already stated in the Results.

Reviewer 2 Report

The article is clear, the study was precisely done and the results are presented in detail, but the statistical analysis remained at a descriptive level only, in fact.
Authors describe the differences among the measurement methods
using distinct accuracy metrics (2) - (7).
But there is no statistical proof, that any of the methods is significantly better than the others, except for the mention of the Wilcoxon test (line 351), but there are no appropriate p-values in the text.
I would expect any model to test, whether some of the factors (e.g. the resolution, the ground determination method or the seedling size) significantly affects the accuracy (like for instance the logistic regression model, where the outcome could be the truly estimated height, with given precision).
Last but not least, I do not see the statistical meaning of calculating the average rank of the six different metrics, as presented in the Table S1_1; this rank can not be interpreted as the "proof of the method quality".
I would recommend shortening the amount of descriptive characteristics and performing one thorough analysis using statistical testing approach.
